Lower-limb sagittal joint angles during gait can be predicted based on foot acceleration and angular velocity

http://orcid.org/0000-0001-8806-7389 Inai Takuma 1 takuma.inai@aist.go.jp
Takabayashi Tomoya 2
1 National Institute of Advanced Industrial Science and Technology , Takamatsu City , Japan
2 Niigata University of Health and Welfare , Niigata City , Japan
Vieira Marcus
Electronic publication date: 2023 Sep 18
Publication date: 2023
Volume: 11
Electronic Location ID: e16131
Received 2023 May 26; Accepted 2023 Aug 28
Copyright: © 2023 Inai and Takabayashi
Copyright year: 2023
Copyright holder: Inai and Takabayashi
License: This is an open access article distributed under the terms of the Creative Commons Attribution License, which permits unrestricted use, distribution, reproduction and adaptation in any medium and for any purpose provided that it is properly attributed. For attribution, the original author(s), title, publication source (PeerJ) and either DOI or URL of the article must be cited.
License URL: https://creativecommons.org/licenses/by/4.0/

Keywords: Gait, Walking, Lower-limb sagittal joint angle, Acceleration, Angular velocity, Foot

Funding: Grant-in-Aid for Young Scientists JP22K17583 Japan Society for the Promotion of Science This study was supported by a Grant-in-Aid for Young Scientists (JP22K17583) of the Japan Society for the Promotion of Science. The funders had no role in study design, data collection and analysis, decision to publish, or preparation of the manuscript.

==============================
Background and purpose

Continuous monitoring of lower-limb movement may help in the early detection and control/reduction of diseases (such as the progression of orthopedic diseases) by applying suitable interventions. Therefore, it is invaluable to calculate the lower-limb movement (sagittal joint angles) while walking daily for continuous evaluation of such risks. Although cameras in a motion capture system are necessary for calculating lower-limb sagittal joint angles during gait, the method is unrealistic considering the setting is difficult to achieve in daily life. Therefore, the estimation of lower-limb sagittal joint angles during walking based on variables, which can be measured using wearable sensors (e.g., foot acceleration and angular velocity), is important. This study estimates the lower-limb sagittal joint angles during gait from the norms of foot acceleration and angular velocity using machine learning and validates the accuracy of the estimated joint angles with those obtained using a motion capture system.

Methods

Healthy adults (n = 200) were asked to walk at a comfortable speed (10 trials), and their lower-limb sagittal joint angles, foot accelerations, and angular velocities were obtained. Using these variables, we established a feedforward neural network and estimated the lower-limb sagittal joint angles.

Results

The average root mean squared errors of the lower-limb sagittal joint angles during gait ranged between 2.5°–7.0° (hip: 7.0°; knee: 4.0°; and ankle: 2.5°).

Conclusion

These results show that we can estimate the lower-limb sagittal joint angles during gait using only the norms of foot acceleration and angular velocity, which can help calculate the lower-limb sagittal joint angles during daily walking.

Introduction

The lower-limb sagittal joint angle is an important variable for evaluating the risk of progression of hip and knee osteoarthritis (Hatfield, Stanish & Hubley-Kozey, 2015; Chang et al., 2017; Kumar et al., 2018). Kumar et al. (2018) reported that a higher hip flexion angle and lower hip extension angle during the early and late stance phase, respectively, were potential risk factors for the progression of hip osteoarthritis. Hatfield, Stanish & Hubley-Kozey (2015) reported that the difference between the peak ankle dorsiflexion angle during the stance phase and the peak ankle plantar flexion angle during the swing phase increased the risk of knee arthroplasty. Considering the progression of these diseases can decrease the quality of life (Salaffi et al., 2005), it is important to evaluate and control the risks. To continuously evaluate the risks of progression of these orthopedic diseases, it is necessary to quantify the lower-limb sagittal joint angles during daily walking.

There are several methods for calculating the lower-limb sagittal joint angles during gait: (1) a motion capture system that includes cameras with reflective markers (the gold-standard method) (Reznick et al., 2021) and (2) a marker-less motion capture system that includes dedicated cameras, smartphone cameras, or digital cameras (Kanko et al., 2021; Vafadar et al., 2022). However, single or multiple cameras must be set and fixed to employ these methods (Kanko et al., 2021; Reznick et al., 2021; Vafadar et al., 2022). In other words, currently available motion capture systems and digital cameras do not offer portability, and hence, are difficult to use for continuous monitoring of the lower-limb sagittal joint angles in real-world or field settings.

To address this problem, this study estimates the lower-limb sagittal joint angles during gait based on acceleration and angular velocity, which can be measured using an inertial measurement unit (IMU). Although various sensors such as millimeter-wave radar (Alanazi et al., 2022), fiber optics (Mohamed et al., 2012; Kim et al., 2014), electro-goniometer (Rowe et al., 2000; da Silva Camassuti et al., 2015), and e-textiles (Tognetti et al., 2015) can estimate the kinematics according to previous studies, IMUs are portable (light and small), incur low cost, exhibit minimal invasiveness, and are not obtrusive (e.g., full-body suits) (Digo, Pastorelli & Gastaldi, 2022). Owing to these advantages, portable measurement tools can help recruit larger, more diverse populations in clinical studies. Previous studies (Favre et al., 2008, 2009; Cooper et al., 2009; Dorschky et al., 2019; Lim, Kim & Park, 2019; Weygers et al., 2020) have reported methods for estimating lower-limb joint angles using accelerations and/or angular velocities. These studies (Favre et al., 2008, 2009; Cooper et al., 2009; Dorschky et al., 2019; Weygers et al., 2020) used numerous IMUs to calculate the lower-limb joint angles during gait. However, attaching several IMUs to the body is troublesome. Although a previous study (Lim, Kim & Park, 2019) used only one IMU on the lower back to estimate lower-limb sagittal joint angles during gait, repetitive attachment and detachment of the IMU can be problematic in daily life.

Therefore, we address this issue by focusing on “foot” acceleration and angular velocity. Considering a person generally uses shoes to walk outside, we can address the issue assuming the shoes are in use, and hence, can be attached (or embedded) to an IMU. In other words, we may quantify lower-limb sagittal joint angles during gait by wearing shoes during walking and without attaching an IMU to one’s own body (e.g., lower back). Therefore, it is valuable to estimate lower-limb joint angles during gait based on foot acceleration and angular velocity.

Hossain et al. (2022) reported a method for estimating the lower-limb sagittal joint angles during gait using accelerations and angular velocities of the dorsum of the feet ( RMSEs of hip, knee, and ankle were 4.90, 5.44, and 3.76 deg, respectively). However, they used three-axis accelerations and angular velocities (and the norms of three-axis accelerations and angular velocities) of the dorsum of the feet as the input layer of a machine learning model. When a person uses an IMU during daily walking, it is often tilted owing to external impact (e.g., heel contact). In the case of the input layer, including three-axis accelerations and angular velocities, errors of the estimated lower-limb sagittal joint angles may increase owing to the IMU tilts. To address this issue, we use only the norms of the three-axis accelerations and angular velocities for the input layer of a machine learning model considering the norms are not affected by the tilting of the IMU.

Therefore, the purpose of this study is (1) to estimate the lower-limb sagittal joint angles during gait using the norm of foot three-axis accelerations and angular velocities using our previous method (Inai & Takabayashi, 2022) and (2) to confirm the accuracy of the estimated lower-limb sagittal joint angles and compare it with a previous study (Hossain et al., 2022). We expect the accuracy of our model to be similar to the previous study even though our model uses the norm of foot three-axis accelerations and angular velocities as input data.

Methods

Participants

Gait data from 200 healthy community-dwelling adults (male: 100, female: 100, average age: 50.0 (18.9) years (range: 20–77 years), height: 1.63 (0.08) m, body mass: 59.4 (9.6) kg) in the public AIST Gait Database 2019 (Kobayashi et al., 2019) were used for this study. The inclusion criteria for analysis were as follows: (1) poses the ability to walk independently without a walking aid; (2) no orthopedic or neurological diseases; (3) no pain in the lower limbs; (4) had normal or corrected-to-normal vision; and (5) age greater than 20 years. The experimental protocol of the dataset was approved by the ethics committee of the National Institute of Advanced Industrial Science and Technology (IRB number: 71120030-E−20150303-002). All participants provided written informed consent before participating in the experiment.

Experiment

Kobayashi et al. (2019), who established the public AIST Gait Database 2019, used a motion capture system (Vicon, Oxford, UK) at a sampling frequency of 200 Hz and six force plates (AMTI, Watertown, MA, USA) at a sampling frequency of 1,000 Hz. An experimenter attached 55 reflective markers to the body of each participant (Kobayashi et al., 2019). All participants were asked to walk barefoot on a straight 10 m path in the laboratory at a self-selected speed. We performed ten successful trials (five trials for each leg) for each subject.

Data analysis

Data preprocessing

Raw marker trajectory data were filtered using a fourth-order Butterworth filter with zero lag and a cutoff frequency of 10 Hz. The raw data of the ground reaction forces (GRFs) were filtered with zero lag and a cutoff frequency of 56 Hz (van den Bogert & de Koning, 1996). We analyzed the gait cycle from heel contact to the next ipsilateral heel contact. The heel contact timing was determined using the vertical GRF.

Local coordinate system

We generated a local coordinate system each for the pelvis, thighs, and shanks using the same method as that in (Kobayashi et al., 2019). The reference was set the long axis of the foot segment from the proximal point (midpoint of the lateral and medial malleolus) to the distal point (midpoint of the first and fifth metatarsal heads). However, in this case, an inappropriate ankle sagittal joint angle was obtained. Therefore, we created a local coordinate system for the left (and right) foot to change the setting of the long axis of the foot segment. (1) Origin: The midpoint of a line connecting the left heel marker and the midpoint of the line connecting the left first and fifth metatarsal head markers.

(2) x-axis (right/left): The unit vector is the cross product of the y- and z-axis.

(3) y-axis (anterior/posterior): Unit vector from the left heel marker to the midpoint of the line connecting the left first and fifth metatarsal head markers.

(4) z-axis (superior/inferior): Unit vector, which is the cross product of a vector from the left heel marker to the first metatarsal head marker and y-axis.

Calculations of joint angles, acceleration, and angular velocity

We calculated the ipsilateral hip, knee, and ankle sagittal joint angles (Cardan angle [sequence: x-y-z]) using the local coordinate systems of the pelvis, ipsilateral thigh, ipsilateral shank, and ipsilateral foot. We calculated the three-axis accelerations of a virtual foot marker (midpoint of “midpoint of the first and fifth metatarsal heads” and “midpoint of the medial malleolus and lateral malleolus,” as shown in Fig. 1 (Delp et al., 2007; Seth et al., 2018)) for each foot during gait using the central difference method. Subsequently, we calculated the norm of the three-axis acceleration for each foot (Fig. 1). The three-axis angular velocities for each foot’s local coordinate system during gait were calculated using the quaternion. The quaternion in each foot segment during gait was in turn calculated using Rodrigues’ rotation formula (Wolfram MathWorld, 2023). Thereafter, we calculated the norm of the three-axis angular velocity for each foot (Fig. 1). The lower limb sagittal joint angles in the ipsilateral lower limb, norm of three-axis accelerations, and norm of three-axis angular velocities during gait were time-normalized to 200 points.

Figure 1 (A–E) Explanation of the virtual foot markers and norms of foot acceleration and angular velocity during gait.

The red circle denotes the midpoint of “midpoint of the first and fifth metatarsal heads” and “midpoint of the medial malleolus and lateral malleolus.” MML, medial malleolus; LML, lateral malleolus; MT1, first metatarsal head; MT5, fifth metatarsal head. Additionally, the lower-limb sagittal joint angles in the ipsilateral limb during gait are used as inputs for the FNN model. We analyze a gait cycle of ipsilateral limb (e.g., if the ipsilateral limb is on the left side, a gait cycle is from the left heel contact to the next left heel contact), and contralateral limb means opposite side. Part A was created using OpenSim software (Delp et al., 2007; Seth et al., 2018).

Method to estimate lower-limb sagittal joint angles during gait

Previous studies (Lim, Kim & Park, 2019; Mundt et al., 2020) used the feedforward neural network (FNN) and successfully estimate the lower-limb joint angles during gait with sufficiently high accuracy (e.g., the RMSE for the joint angle prediction using five sensors was smaller than 4.8° (Mundt et al., 2020)). As a result, it can potentially exhibit a highly accurate estimation with FNN. Therefore, we also used the FNN (Inai & Takabayashi, 2022) to estimate the lower-limb sagittal joint angles during gait.

As shown in Fig. 2, the gait data of all participants (foot acceleration, angular velocity matrix, and joint angle matrix) were randomly divided into five groups: (1) three groups for training data; (2) one group for validation data; and (3) one group for test data. Additionally, data from a subject was able to be present in one of these groups (i.e., 10 trials in a subject are included in one of these groups, and the data was divided at a subject level when randomizing the folds).

Figure 2 Data split for the input of our FNN model.

(A and B) Data for the input and output layers, respectively. The green, blue, red, and yellow indicate the validation data, training data, test data, and a foot acceleration and angular velocity (FAAV) matrix of a subject, respectively. The data is randomly divided into five groups, and 10 trials of the same subject are placed in the same group.

We performed hyperparameter tuning to determine the optimal parameters for the main model (see “Hyperparameter Tuning” section for details). Subsequently, the test data were used as input to the main model to confirm the accuracies of the estimated lower-limb sagittal joint angles during gait.

Hyperparameter tuning

We set 324 conditions (=12 [number of hidden layers and nodes in each hidden layer] × 3 [batch size] × 3 [dropout rate] × 3 [learning rate]) (see Appendix A for details). We used k-fold cross-validation to acquire robust hyperparameters (Kawamoto & Kabashima, 2017). Figure 3 shows a flowchart of the k-fold cross-validation (k = 4 in this study) for hyperparameter tuning (e.g., Dataset2).

Figure 3 Flowchart of the k-fold cross-validation for hyperparameter tuning.

FAAV, foot acceleration and angular velocity; FNN, feedforward neural network; PCA, principal component analysis; PCL, principal component loading; PCS, principal component score.

The foot accelerations and foot angular velocities at each gait cycle point may have highly correlated features during walking, resulting in an unstable FNN model. To address this issue, we hypothesized that principal component analysis (PCA) is an appropriate approach considering it can reduce the dimension of the data (Kobayashi et al., 2014, 2016; Tsuchida et al., 2022). Therefore, we performed PCA for the foot acceleration, angular velocity matrix (1,200 × 800), and joint angle matrix (1,200 × 600) of the training data to learn the parameters of a linear transformation from the training data. Thereafter, the transformation was applied to the validation and test data using the learned parameters.

We obtained the principal component loading (PCL) matrix for input (800 × 65; MPCL_input_D2), principal component score (PCS) matrix of the input training data (1,200 × 65; MPCS_input_D2), PCL matrix for the output (600 × 17; MPCL_output_D2), and the PCS matrix of the output training data (1,200 × 17; MPCS_output_D2) using PCA. Additionally, we obtained principal component vectors (PCVs) with eigenvalues of one or more in the PCA (i.e., 65 and 17 for the input and output layers, respectively). We used the PCS matrix of the training data for the input (MPCS_input_D2) as the input layer. Similarly, we used the PCS matrix of the training data for the output (MPCS_output_D2) as the output layer as the ground truth.

The relationship between the waveform and PCS can be expressed as:

(1) MWaveformT=MMean+MSDMPCLMPCST

(2) MMean=[m1,1⋯m1,1⋮⋱⋮md,1⋯md,1]

(3) MSD=[σ1⋯0⋮⋱⋮0⋯σd]

(4) MPCL=[x1,1⋯x1,n⋮⋱⋮xd,1⋯xd,n],and

(5) MPCS=[y1,1⋯y1,n⋮⋱⋮yt,1⋯yt,n]

where MWaveform, MMean, MSD, MPCL, and MPCS denote a t×d matrix of waveforms, d×t matrix including the mean value m for each point, d×d diagonal matrix including the standard deviation (SD) σ for each point, d×n matrix including PCL x for each point and PCV, and t×n matrix including PCS y for each PCV, respectively. t denotes the number of trials, d denotes the number of data points, and n denotes the number of PCVs. We used the PCL matrix for the input (MPCL_input_D2) as well as the foot acceleration and angular velocity matrix of the validation data to calculate the PCS matrix of the validation data for the input (MPCS_input_D2_val). Subsequently, we used a PCS matrix for the input layer.

We calculated the mean absolute errors between the actual PCS matrix of the validation data for the output and the predicted PCS matrix of the validation data for the output in 1,296 patterns (four datasets (i.e., Dataset1–Dataset4) × 324 conditions), and the mean absolute errors were averaged under each condition. Furthermore, we determined the optimal parameters (number of hidden layers, nods for each hidden layer, batch size, drop rate, and learning rate) under the condition with the lowest mean absolute errors for the optimal parameters of the main model.

We used the Swish function (Ramachandran, Zoph & Le, 2017) as the activation function. The number of epochs was set to 300. Adaptive moment estimation was used as the optimizer (Kingma & Ba, 2017).

Main model

Figure 4A shows the flowchart of the main model (training). We performed PCA for the foot acceleration and angular velocity matrix (1,600 × 800), as well as the joint angle matrix (1,600 × 600) of the training data to reduce the dimensions of the data. We obtained the PCL matrix for the input (800 × 65; MPCL_input_main), PCS matrix of training data for the input (1,600 × 17; MPCS_input_train), PCL matrix for the output (600 × 17; MPCL_output_main), and PCS matrix of training data for the output (1,600 × 17; MPCS_ output _train) using PCA.

Figure 4 Flowchart of the main model.

(A) For training and (B) for testing.

Figure 4B shows the flowchart of the main model (test). We calculated the PCS matrix of the test data for input (400 × 65; MPCS_input_test) using the foot acceleration and angular velocity matrix of the test data (400 × 800) and the PCL matrix for input (800 × 65; MPCL_input_main) using Eqs. (1)–(5). We input the PCS matrix of the test data for the input layer and calculated the PCS matrix of the output test data (400 × 17; MPCS_output_test). We calculated the predicted joint angle matrix (400 × 600) using the PCS matrix of the output test data (400 × 17; MPCS_output_test) and the PCL matrix for the output (600 × 17; MPCL_output_main) using Eqs. (1)–(5).

We evaluated the accuracy of the predicted lower limb sagittal joint angles during gait (Fig. 4B) using the average root-mean-square error ( RMSE¯) (deg), normalized root-mean-square error ( NRMSE¯) (%), and Pearson correlation coefficient ( ρ¯) based on a previous study (Sivakumar et al., 2021). Furthermore, we calculated the mean absolute values using the actual and estimated lower-limb joint angles at various timings (hip flexion angle at heel contact, peak hip extension angle, peak knee flexion angle, ankle dorsiflexion angle at heel contact, peak ankle dorsiflexion angle, and peak ankle plantar flexion angle).

Results

Hyperparameter tuning

The optimal number of hidden layers, nodes in the hidden layer, batch size, drop rate, and learning rate were 1, 80, 512, 0.5, and 0.001, respectively (see Appendix A for details).

Main model

Table 1 lists the accuracies of the proposed method and those of a previous study (Hossain et al., 2022). Figure 5 shows the actual and estimated hip, knee, and ankle joint angle waveforms during the gait. It can be seen that the actual and predicted lower-limb joint angle waveforms were extremely similar. Table 2 lists the mean absolute errors of joint angles at various timings.

Table 1 Comparison between the proposed method and a previous study (Hossain et al., 2022).

		Hip joint angle	Knee joint angle	Ankle joint angle	
	n	RMSE (°)	NRMSE (%)	ρ	RMSE (°)	NRMSE (%)	ρ	RMSE (°)	NRMSE (%)	ρ	
Method by Hossain et al.	10	4.9	–	0.96	5.4	–	0.98	3.8	–	0.96	
Proposed method	188	7.0	14.2	1.00	4.0	6.5	0.99	2.5	7.9	0.98	

Figure 5 Lower limb sagittal joint angles during gait.

(A–C) Hip, knee, and ankle sagittal joint angles, respectively. The red solid and black dashed lines denote the actual and predicted waveforms (test data), respectively.

Table 2 Mean absolute errors for each joint angle (mean (SD); °).

Hip flexion angle at heel contact	6.2	(4.7)	
Peak hip extension angle	7.5	(5.1)	
Peak knee flexion angle	2.7	(2.1)	
Ankle dorsiflexion angle at heel contact	2.1	(1.6)	
Peak ankle dorsiflexion angle	1.8	(1.4)	
Peak ankle plantar flexion angle	2.9	(2.2)	

Discussion

We estimated the lower-limb sagittal joint angles during gait using the norms of acceleration and angular velocity for each foot and confirmed the accuracy of the predicted lower limb joint angles during gait. The results demonstrated that our method can estimate the lower-limb sagittal joint angles using foot acceleration and angular velocity with accuracies of RMSEs¯ of 2.5°–7.0° (Table 1).

The accuracies of the estimated knee and ankle sagittal joint angles during gait were better compared to those of previous studies (Hossain et al., 2022) ( RMSEs¯ of the knee and ankle joint angles in the present study were 4.0° and 2.5°, respectively, and those in the previous study were 5.4° and 3.8°; Table 1). However, the accuracy of the estimated hip sagittal joint angle during gait was lower than that in a previous study (Hossain et al., 2022) ( RMSE¯ of the hip joint angle in the present and previous study was 7.0° and 4.9°, respectively; Table 1), considering the type of machine learning (Hossain et al., 2022) and (2) the input data (Hossain et al., 2022) of the present and previous studies were different. First, we used a general FNN with hidden layers and nodes to estimate the lower-limb sagittal joint angles during gait. Conversely, previous studies used a framework (DeepBBWAE-Net) by implementing bagging, boosting, and weighted average ensemble techniques to estimate lower-limb sagittal joint angles during various movements, including overground walking (Hossain et al., 2022). Furthermore, they stated that DeepBBWAE-Net could estimate joint kinematics more accurately than other simpler deep-learning models (Hossain et al., 2022). Additionally, although the inputs for our machine learning technique were the norms of accelerations and angular velocities for each foot, a previous study (Hossain et al., 2022) used three-axis accelerations and angular velocities (including the norm of accelerations and angular velocities for each foot). In other words, the amount of input information for machine learning in the previous study was greater than that in this study. Therefore, the difference in the input for machine learning may affect the results of the estimated hip sagittal joint angles during gait.

According to a previous study (Hatfield, Stanish & Hubley-Kozey, 2015), the difference between the peak ankle dorsiflexion angle during the stance phase and peak ankle plantar flexion angle during the swing phase increases the risk of total knee arthroplasty. Because the accuracy of the estimated ankle (and knee) sagittal joint angle was better (Table 1), the proposed method may be helpful in evaluating the risks in the future. Although our method can estimate the lower-limb sagittal joint angles during gait, the lower-limb sagittal joint angles estimated by our method include slight errors (particularly the hip sagittal joint angle; Table 1). Therefore, the accuracy of the hip angle may not be sufficient to evaluate the risk of orthopedic diseases, such as hip osteoarthritis. For instance, a previous study (Kumar et al., 2018) evaluated the association between lower-limb sagittal joint angles during the stance phase and magnetic resonance imaging changes in the hip joint over 18 months. Consequently, progressors walked with a 4.5° greater hip flexion angle during the early stance phase at baseline (Kumar et al., 2018). In other words, a greater hip flexion angle during the stance phase is associated with a risk of progression to hip osteoarthritis (Kumar et al., 2018). However, RMSE¯ of the hip sagittal joint angle in this study was 7.0° (Table 1). Therefore, we believe our method is beneficial for distinguishing extremely high (or low) hip flexion angles during the stance phase (e.g., hip flexion angle: 45°) from the general hip flexion angle during the stance phase (i.e., hip flexion angle: approximately 23°). However, if a person has a slightly higher hip flexion angle during the stance phase (e.g., hip flexion angle: 28°), it can be difficult to evaluate the risk of orthopedic diseases accurately. Therefore, in the future, we may need to improve our estimation method (or propose a new method) to increase the accuracy when evaluating the risk of orthopedic diseases.

This study had some limitations. First, we used the walking data of healthy adults as the input for machine learning. Owing to the differences in the lower-limb sagittal joint angles during gait between healthy adults and patients with hip or knee osteoarthritis (Eitzen et al., 2012; Bytyqi et al., 2014), the proposed model may not be able to accurately estimate the lower-limb sagittal joint angles during gait in participants with orthopedic diseases. Therefore, future studies should recruit patients with orthopedic diseases and confirm the accuracy of lower-limb sagittal joint angles during gait. Second, we only used laboratory data on straight walking. In general, a person can have a wide range of gait patterns and conditions (e.g., uphill and downhill walking, turning, and various walking speeds). Therefore, future studies should acquire a comprehensive dataset, such as a previous study (Warmerdam et al., 2022), that encompasses a wide range of gait patterns and conditions to establish a more valid estimation model. Third, acceleration and angular velocity by an actual IMU were not used. Because there is a difference between the data of an IMU and that of a motion capture system (Grimmer et al., 2019), acceleration and angular velocity by an actual IMU were not used to avoid the inaccuracy of prediction by the model. Therefore, we must conduct experiments by combining IMU and a motion capture system to update the estimation model that uses the data of IMUs in the future. Finally, we used only machine learning of the feedforward neural network model. There are some types of machine learning such as long-short term memory, convolutional neural network, and DeepBBWAE-Net (Hossain et al., 2022). We believe that we should examine the type of machine learning technique that can estimate lower-limb sagittal joint angles using the norms of accelerations and angular velocities.

Conclusions

This study estimated the hip, knee, and ankle sagittal joint angles during gait using the norms of acceleration and angular velocity for each foot and confirmed the accuracy. To the best of our knowledge, this is the first study to estimate lower-limb joint angles using norms of foot acceleration and angular velocity during gait. Although the norms of foot acceleration and angular velocity are not affected by errors of sensor tilt, the results showed that the proposed method can estimate lower limb sagittal joint angles with an RMSEs¯ accuracy between 2.5°–7.0°.

Supplemental Information

Supplemental Information 1 Combinations of number of hidden layers, number of nodes in hidden layers, batch size, drop rate, and learning rate.

The yellow highlighted letters indicate optimal hyperparameters in the present study.

Click here for additional data file.

Supplemental Information 2 Main code (R language).

Click here for additional data file.

Additional Information and Declarations

Competing Interests

Author Contributions

Human Ethics

Data Availability

The authors declare that they have no competing interests.

Takuma Inai conceived and designed the experiments, analyzed the data, prepared figures and/or tables, authored or reviewed drafts of the article, and approved the final draft.

Tomoya Takabayashi conceived and designed the experiments, prepared figures and/or tables, authored or reviewed drafts of the article, and approved the final draft.

The following information was supplied relating to ethical approvals (i.e., approving body and any reference numbers):

Advanced Industrial Science and Technology granted Ethical approval to carry out the study within its facilities (IRB number: 71120030-E−20150303-002).

The following information was supplied regarding data availability:

The code is available in the Supplemental File.

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
