# Peer review of "Lower-limb sagittal joint angles during gait can be predicted based on foot acceleration and angular velocity"

_PeerJ, doi:10.7717/peerj.16131_

## Round 0.1 · original submission · Major Revisions

· Academic Editor

Major Revisions

Please, consider carefully the comments of both Reviewers, especially that of Dr. Hansen. As Dr. Hansen has commented, the authors should consider getting real IMU data and test other machine learning algorithms to confirm FNN is the best. Besides, it would be desible to make the software freely accesible.

Reviewer 1 ·

Basic reporting

Overall, the article is well written, and the analysis methods are thoroughly described. Findings of the study state that the demonstrated method was able to estimate knee and ankle joint angles with better accuracy. Below is my critical review of the article in the form of constructive comments to the authors, which may be beneficial in further improving the quality of the manuscript prior to acceptance.

1.1. The abstract is structured. The authors may be more specific on how continuous monitoring may be helpful. For instance, on Line 18 (Abstract), it can be explicitly stated that continuous monitoring of lower-limb movement may help in early detection of diseases and ultimately, control/reduction in risk by applying suitable interventions.

1.2. On line 27 (Abstract), the statement ‘confirmed accuracies using the estimated joint angles with actual joint angles (i.e., a motion capture system).’ can be made clearer as: ‘validated the accuracy of estimated joint angles with those obtained using a motion capture system’.

1.3. Line 57: The authors can rephrase to include the word ‘portability’ and may state that currently available motion capture systems and digital cameras do not offer portability, which makes them difficult to be applied for continuous monitoring in real-world or field settings.

1.4. Line 85: Did any of the previous studies on estimating lower-limb sagittal joint angles validate their outcomes using a motion capture system? If so, please also state their accuracies and then mention that one of the potential reasons for their reduced accuracy could be that they did not use norms of acceleration and velocity.

1.5. The last paragraph of introduction Page 3 Line 95 may be improved by including an ending statement about the broad implications and benefits of this study. Please also state the hypothesis, or expectations (that it is expected that including norms as input may lead to higher accuracy).

1.6. Figure 1: Please indicate the meaning of terms ‘ipsilateral’ and ‘contralateral’ for readers who are unfamiliar with the terminology. This can be added either in the figure captions or in the methods section.

1.7. Table 1: Please provide more detailed caption such as: ‘Comparison between the proposed method and a previous study (Hossain et al.)’

Experimental design

2.1 The goals of this study are well defined, and align with the scope of this journal.

2.2 Line 161 ‘sufficiently high accuracy’: If possible, please be more specific and include the reported values for accuracy.

2.3 Overall, the analysis methods implemented in this study are thoroughly explained.

Validity of the findings

3.1 Line 260: Although the accuracies have been presented in the table, it may be helpful to include the improvement in writing here as well. For instance, how much % improvement was observed?

3.2 Line 277: Please mention here that one of the limitations of the implemented methods in this study is the lower accuracy of hip sagittal joint angle.

3.3 Line 278: I commend the authors for including a paragraph stating that the method in its current state may not be applicable for detecting slight changes in hip angle to detect the risk of orthopedic diseases. However, there may be other applications (where foot and knee sagittal angles may be crucial) that the authors may consider where the proposed method may be more beneficial compared to currently used methods. Besides continuous monitoring, the demonstrated method may also be beneficial in studies related to analyzing other health conditions, as an in-field assessment tool, in rehabilitation procedures, or in the estimation of joint loads (as a more accessible input due to reduced number of wearable sensors for inverse dynamics method). I encourage the authors to find relevant studies in these areas and improve the discussion section of the manuscript.

3.4 Line 319: The authors may state the accuracies for hip, knee, and ankle separately, and mention that the accuracies were better for the knee and ankle joint angles but were less accurate for hip joint.

Additional comments

4.1 The font color of the axes of the graphs in Figure 1 and Figure 5 may be changed from grey to black to improve the quality.

·

Basic reporting

The study titled "Lower-limb sagittal joint angles during gait can be predicted based on foot acceleration and angular velocity" presents a novel approach to predicting lower-limb sagittal joint angles using machine learning models. While the concept is intriguing and has the potential to provide valuable insights into gait analysis, it is crucial to address certain limitations and question the choice of the machine learning model and explain the clinical reasoning for estimating sagittal joint angles (line 50-52). However, the overall writing style of the article could benefit from significant improvement. Sometimes the structure is not entirely clear and some paragraphs are incoherent. Out of the blue, the authors compare their results with a previously reported method from Hossain et al 2022, which has not been introduced in the objectives.

Experimental design

Would it also be possible e.g. to use other machine learning models models? The authors base their methods (L 158-163) solely on three articles mentioning sufficient high accuracy, so the choice is not very well established.
One notable limitation of the study is the absence of real IMU data. To address this limitation, a comparison with real IMU data and the obtained data from double differentiation (from filtered 10Hz data) can provide more reliable and robust information and would significantly improve the accuracy and consequently enhance the predictions of lower-limb sagittal joint angles during gait.

Validity of the findings

One of the primary concerns with this study is the limited availability of clinical data. The absence of an extensive and diverse dataset derived from a representative population raises doubts about the reliability and generalizability of the machine learning model's predictions. A more robust and comprehensive dataset, encompassing a wide range of gait patterns and conditions, is imperative to establish the model's validity and practical utility. We propose to have a look at e.g. https://www.mdpi.com/2306-5729/7/10/136, especially with the home-like assessments. The authors express their wish to use their models during home assessment e.g. when participants wear IMUs over the course of a few days in their home.

Additional comments

One additional aspect that warrants attention is the absence of open-source data and analysis software. The unavailability of open-source software and data hinders the transparency and reproducibility of the study's findings, limiting the ability of other researchers to validate and build upon the proposed predictive model. Furthermore, it restricts the dissemination of the methodology, hindering potential advancements in the field of gait analysis. Addressing these limitations by making data and analysis software openly accessible would significantly enhance the study's impact and foster collaboration and advancements within the research community.

---

## Round 0.2 · Minor Revisions

· Academic Editor

Minor Revisions

I deeply appreciated the comments of both reviewers who rigorously reviewed the manuscript. Based on those comments, the authors have noticeably improved the article. I think the manuscript is practically ready to be published. I recommend, however, to consider the minor comments of Reviewer 1.

Reviewer 1 ·

Basic reporting

1.1. The changes made in the revised manuscript for my earlier comments are satisfactory. I do have some concerns about the novelty of this study since taking the norm of three-axis acceleration is quite common in studies involving motion analysis by implementing inertial sensors. This is especially true for segmenting tasks involving complex body movements. The authors have however validated the accuracy of the method using a large sample size (200 healthy adults), and the findings may be of value to future studies.

1.2. Please explicitly state the assumptions made in this study in the form of paragraph in either the methodology, or discussion section. For instance, potential differences arising from using real-world IMU data vs. the data from motion capture system used in this study. IMU-data is subject to drift errors. Can these measurement errors lead to any differences in the outputs of the model? Please state your expectations, by referring to previous literature on this topic.

Experimental design

no comment

Validity of the findings

The response letter, and the changes made in the revised manuscript based on my earlier comments are satisfactory.

·

Basic reporting

The revised manuscript submitted by the authors showcases notable improvements. The article maintains a consistent use of clear English throughout. Adequate references to relevant literature and ample field background and context contribute to the article's scholarly value. The organization of the article, along with the well-prepared figures and tables, facilitates effective information conveyance. The authors' sharing of the code is a positive step towards transparency. Furthermore, the manuscript remains self-contained, offering relevant results that align with the stated hypotheses. These revisions collectively reflect the authors' commitment to producing a substantive contribution to the field.

Experimental design

The revised manuscript presents itself as a substantial contribution within the defined Aims and Scope of the journal. The research question is appropriately formulated, displaying relevance and significance, and the authors clarify how their work addresses a specific knowledge gap. The investigation undertaken is marked by rigorous adherence to both technical and ethical standards. The description of methods employed provides adequate detail for potential replication.

Validity of the findings

The revised manuscript presents notable improvements and exhibits several commendable attributes. The evaluation of impact and novelty is well-addressed, highlighting potential avenues for future research. The encouragement of meaningful replications, when substantiated by a clear rationale and benefits to the existing literature, adds depth to the manuscript's significance. The provision of all underlying data is noteworthy, attesting to their robustness, statistical validity, and careful control. The conclusions drawn are aptly tied to the initial research question and thoughtfully limited to supporting the presented results. These revisions collectively enhance the manuscript's scientific value.

Additional comments

no additional comments

---

## Round 0.3 · accepted · Accept

· Academic Editor

Accept

The authors have addressed all of the reviewers' comments. The manuscript has improved substantially and it is ready to be published. Thanks to the Reviewers and authors.

Reviewer 1 ·

Basic reporting

The authors have made several improvements to the draft and I appreciate the authors for considering my comments. I am satisfied with the current draft and the believe that the paper can be published in its current form.

Experimental design

no comment

Validity of the findings

no comment